:ᐧ☼ᐧ PLOS | ONE

# Long-term vancomycin use had low risk of ototoxicity

**Clayton Humphrey[1], Michael P. Veve ᴵᴰ[1,2], Brian Walker[1], Mahmoud A. Shorman ᴵᴰ[1]***

**1** University of Tennessee Graduate School of Medicine, Knoxville, Tennessee, United States of America,
**2** University of Tennessee Health Sciences Center, Knoxville, Tennessee, United States of America

* mshorman@utmck.edu

## Abstract

### Background

Vancomycin is a commonly used antibiotic with potent activity against Gram-positive organisms, but prolonged use and high doses can lead to toxicity. While vancomycin-associated nephrotoxicity is widely reported, few cases of ototoxicity have been described. The objective of this study was to determine the prevalence of negative changes in audiograms in patients receiving long-term intravenous (IV) vancomycin and to identify high-risk patients who need audiogram monitoring.

### Methods

This was an IRB approved, cross-sectional study performed at an academic medical center from 1/2012-3/2019. Patients who were prescribed IV vancomycin for $\geq$ 14 days and had baseline and follow-up weekly audiometry were included. All data was extracted from the electronic medical record. The primary endpoint was worsening audiogram while on vancomycin. Descriptive and bivariate statistics were used to describe the patient population.

### Results

424 patients were screened for inclusion; 92 received at least two audiograms while on vancomycin. Fifty-three percent of patients were men, the median (IQR) patient age was 44 (34–58) years, and 8% of patients had an estimated Cockcroft-Gault creatinine clearance $\leq$ 30 mL/min or received hemodialysis. The median (IQR) vancomycin exposure up until the last recorded audiogram was 30 (17–42) days. Vancomycin indications were: 53 (58%) bone and joint infections, 17 (18%) infective endocarditis, 10 (11%) bacteremia, 12 (13%) other infections. Seven (8%) patients experienced a worsening change in hearing from baseline, two (2%) of them suffered mild loss, two (2%) had mild to moderate loss, and three (3%) developed moderate-to-severe hearing loss. In bivariate analyses, no variables were found to be associated with a worsening change in audiogram, including baseline abnormal audiogram, age $\geq$ 40 years, elevated serum vancomycin levels, or vancomycin doses $\geq$ 4 grams/day.

**Data Availability Statement:** All relevant data are within the paper and its Supporting Information files.

**Funding:** The authors received no specific funding for this work.

**Competing interests:** The authors have declared that no competing interests exist.

## Conclusions

The prevalence of negative changes in audiograms among patients receiving long-term intravenous vancomycin was low. The utility of routine audiogram testing in this population remains questionable except in high-risk patients; however, larger prospective studies with controls may be warranted to further explore the risk of ototoxicity.

## Background

Vancomycin, one of the most widely used antibiotics in the United States, is a glycopeptide antibiotic that has been in clinical use since 1954. It is mainly used for the treatment of severe Gram-positive infections, such as methicillin-resistant *Staphylococcus aureus* (MRSA) [1–3]. While initial vancomycin utilization was limited due to formulation impurity issues and a low prevalence of invasive MRSA infections, recent usage has had a dramatic increase due to the commonality of community and healthcare-associated MRSA infections [4–5].

Increasing vancomycin dosages are often needed to account for elevated vancomycin minimal inhibitory concentrations (MICs) observed in MRSA isolates, which are thought to be related to treatment failure among patients with bacteremia and pneumonia, especially in obese patients [6–7]. Nephrotoxicity is a relatively common adverse effect associated with prolonged vancomycin use, with new reports suggesting a higher incidence of vancomycin-associated nephrotoxicity when larger doses are used to maintain therapeutic efficacy for trough-based goals [8].

Ototoxicity is a less frequently described adverse effect of vancomycin and has not been as readily demonstrated, however vestibular damage and/or cochlear damage associated with tinnitus and sensorineural hearing loss has been reported in humans after administration of vancomycin [9–14]. The use of other ototoxic drugs known to cause sensorineural hearing loss, such as aminoglycosides or loop diuretics, concomitantly with vancomycin makes it difficult to prove a causal association [15].

The 2009 therapeutic monitoring guidelines for vancomycin in adults do not recommend routine audiogram monitoring for ototoxicity [1], but rather to consider the ototoxic potential of the drug in patients receiving additional ototoxic medications [1,16]. Questions remain related to the true vancomycin ototoxicity risk potential, particularly in the setting of more aggressive vancomycin dosing and in the aging population seen in clinical practice. Monitoring ototoxicity with serial audiograms is costly and patient compliance may be a barrier to routine monitoring [17]. Given the perceived risks of vancomycin- associated ototoxicity and the commonality of invasive MRSA infections in our region, audiogram monitoring for patients receiving prolonged vancomycin for any indication was adapted by select infectious diseases physicians at our institution.

The objective of this study was to evaluate the prevalence of and identify risk factors for worsening audiogram changes in patients receiving long-term intravenous vancomycin. Additionally, we sought to determine the need for the serial audiogram monitoring from an antimicrobial stewardship prospective.

## Methods

This was a cross-sectional study performed at an academic medical center from January 2012 and March 2019. A nested case control analysis was conducted on patients who developed

changes in audiograms. Cases included patients who developed a worsening change in follow-up audiogram, controls had no change in follow-up audiogram. Patients were included only once in the study to preserve statistical independence. The study was approved by the University of Tennessee Graduate School of Medicine institutional review board, and a waiver of informed consent was obtained.

Data were retrospectively gathered using chart review from the electronic medical record. Patients were included if they were $\geq$ 18 years old, prescribed vancomycin for $\geq$ 14 days, and had a baseline (within 3 days of initiating vancomycin therapy) and weekly follow-up audiograms. Patients were excluded if they had dementia or other cognitive dysfunction, a documented history of perforated tympanic membrane or the presence of hearing aids or cochlear implants, or if they did not have any follow-up audiograms.

Medical records were reviewed after patient identification and then standardized per protocol, and data collected included the following: patient age, race, sex, baseline serum creatinine and estimated creatinine clearance using the Cockroft-Gault equation, vancomycin dosing regimen used (i.e., dose, frequency, duration), all available vancomycin serum concentrations, infection type and vancomycin indication, microbiology and isolated organism MIC, and concomitant use of other ototoxic medications (i.e., aminoglycosides, furosemide and platinum-based chemotherapy).

The primary endpoint was worsening change in audiogram while on vancomycin. The procedure for audiogram monitoring in patients receiving vancomycin was not formally protocolized, but patients thought to be at risk for ototoxicity by the ordering provider typically received orders for weekly audiograms throughout the duration of vancomycin. Baseline and follow-up audiogram results were collected; certified audiologists interpreted audiograms. Findings were recorded for both ears ranked in order of increasing severity based on the sound intensity required for hearing as follows: normal, <25 dB; mild, 25 to 40 dB; moderate, 40 to 70 dB; severe, 70 to 90 dB; profound, >90 dB sensorineural hearing loss. Patients were considered to have a negative change in audiogram if there was a noted increase in the sound intensity required for hearing and worsening of more than 5 db from baseline.

Given the exploratory nature of this project, no formal sample size calculations were performed. Descriptive and bivariate analyses were utilized to describe the patient population. Categorical and continuous variables were compared using the Chi-Square, Fisher's Exact, or Mann-Whitney U-tests. Classification and regression tree (CART) analyses were performed to identify dichotomous breakpoints in continuous variables associated with a negative change in audiogram. Rates of missing data for each key variable will be monitored throughout the study with a benchmark of <10% of individuals with missing data for a single key variable. Demographic or potentially confounding variables with missing data in excess of the benchmark will be removed and a suitable proxy variable will be chosen if possible. Patients without confirmed outcome status will be excluded from the analysis of that outcome.

## Results

A list of adult patients who received vancomycin and at least one audiogram or audiogram order placed in the electronic health record was created in order to further screen patients for inclusion. Four-hundred and twenty-four patients were screened for inclusion; 92 (22%) received at least two audiograms while on vancomycin and were treated for at least 14 days. Baseline characteristics of the patient population are listed in Table 1. Most patients were men (53%) and Caucasian (95%); the median (IQR) patient age was 44 (37–61) years. Seven (8%) patients had an estimated creatinine clearance $\leq$ 30 mL/min using the Cockroft-Gault equation; 6 patients received hemodialysis.

**Table 1. Baseline characteristics of patients based on change in audiogram during vancomycin therapy.**

| Variable | Total Population ($n$ = 92) | No change in Audiogram ($n$ = 85) | Negative Change in Audiogram ($n$ = 7) | *P-value*[†] |
|---|---|---|---|---|
| **Patient Demographics** | | | | |
| Median (IQR) Age, years | 44 (34–58) | 43 (33–58) | 43 (38–61) | 0.57 |
| Race, Caucasian | 87 (95%) | 81 (95%) | 6 (86%) | 0.33 |
| Sex, Male | 49 (53%) | 46 (54%) | 3 (43%) | 0.7 |
| Median (IQR) Body Mass Index, kg/m$^2$ | 25.0 (20.5–31) | 25.1 (20.3–30.6) | 23.0 (20.0–37.7) | 0.8 |
| Estimated Creatinine clearance, at time of VAN | | | | |
| >100 mL/min | 46 (50%) | 42 (51%) | 3 (43%) | 1.0 |
| 80–99 mL/min | 9 (10%) | 9 (11%) | 0 | 1.0 |
| 60–79 mL/min | 15 (16%) | 13 (15%) | 2 (29%) | 0.32 |
| 40–59 mL/min | 12 (13%) | 11 (13%) | 1 (14%) | 1.0 |
| 20–39 mL/min | 3 (3%) | 3 (4%) | 0 | 1.0 |
| < 20 mL/min | 7 (8%) | 6 (7%) | 1 (14%) | 0.44 |
| Abnormal audiogram at baseline | 51 (55%) | 47 (56%) | 4 (57%) | 1.0 |
| **Vancomycin Characteristics** | | | | |
| Number of audiograms, Median (IQR) | 3 (2–5) | 3 (2–5) | 2 (2–6) | 0.9 |
| VAN mg/kg per dose, Median (IQR) | 16.8 (14.5–19.2) | 16.9 (14.6–19.4) | 16.2 (13.6–17.9) | 0.54 |
| VAN concentrations, mcg/mL, Median (IQR) | 17.6 (15.4–20.1) | 17.6 (15.5–20.8) | 18.6 (14.5–24.5) | 0.83 |
| VAN serum concentration $\geq$ 20 mcg/mL | 68 (74%) | 64 (75%) | 4 (57%) | 0.37 |
| Daily VAN dose > 4 g/day | 19 (21%) | 17 (20%) | 2 (29%) | 0.63 |
| VAN duration, days, Median (IQR) | 42 (31–42) | 42 (30–42) | 42 (26–56) | 0.82 |
| Concomitant ototoxins | 11 (12%) | 11 (13%) | 0 | 0.59 |
| **Infection Types** | | | | |
| Osteoarticular | 53 (58%) | 47 (55%) | 6 (86%) | 0.12 |
| Endocarditis | 17 (18%) | 17 (20%) | 0 | 0.59 |
| Bacteremia | 10 (11%) | 10 (12%) | 0 | 1.0 |
| Other | 12 (13%) | 11 (13%) | 1 (4%) | 1.0 |

[†]Bivariate comparisons are between patients who did and did not have negative audiogram changes.

Abbreviations: VAN, vancomycin

A total of 351 audiograms were performed for 92 patients; the median (IQR) number of weekly audiograms while on vancomycin and time to first follow-up audiogram was 3 (2–5) and 7 (7–9) days, respectively. The median (IQR) vancomycin exposure up until the last recorded audiogram was 30 (17–42) days. Seventy-two percent of patients received at least 3 or more weekly audiograms while on vancomycin. The initial audiogram was performed while the patient was hospitalized, with subsequent audiograms performed in the outpatient setting for the majority of included patients.

Vancomycin indications were: 53 (58%) bone and joint infections, 17 (18%) infective endocarditis, 10 (11%) bacteremia, 12 (13%) other infections. The median prescribed vancomycin duration was 42 (28–42) days, and all patients had follow-up vancomycin concentrations; the median (IQR) vancomycin concentration was 17.6 (15.4–20.8) mcg/mL, baseline characteristics of patients based on change in audiogram during vancomycin therapy are listed in Table 1.

Seven patients (8%) experienced a worsening change in hearing from baseline audiogram, two patients (2%) experienced mild sensory neural loss, two patients (2%) had mild to moderate loss, and three patients (3%) developed moderate to severe loss. The median (IQR) time to negative audiogram change was 14 (14–28) days. Of the patients with worsening audiograms,

**Table 2. Characteristics of patients with negative audiogram changes.**

| Pt. No. | Age, years | BMI, kg/m² | Estimated CrCl, mL/min | VAN mg/kg/dose | Median VAN trough, mcg/mL | Indication | VAN duration, days | No. Audiograms | Baseline Audiogram Results | Follow-up Audiogram Results | Time to Negative Audiogram Change, days |
|---|---|---|---|---|---|---|---|---|---|---|---|
| 1 | 56 | 33.0 | 71 | 16.75 | 18.33 | Septic Arthritis | 42 | 7.00 | Mild HF HL bilaterally | Mild to sloping severe HL Bilaterally | 28 |
| 2 | 59 | 21.0 | 8 | 14.12 | 20.38 | Osteomyelitis | 40 | 6.00 | Normal Hearing Bilaterally | Mild to Mod HF hearing loss bilateraly | 28 |
| 3 | 45 | 25.0 | 120 | 17.69 | 19.83 | Brain abscess | 48 | 5.00 | Mild LF HL BL, Mod/Severe MidF HL BL | Increased HL MidF Left ear (still mod/severe) | 35 |
| 4 | 41 | 32.0 | 71 | 18.90 | 14.55 | Osteomyelitis | 39 | 4.00 | Mild sloping to at least severe HL bilaterally | Slightly decreased hearing bilaterally from baseline | 14 |
| 5 | 64 | 54.9 | 50 | 12.06 | 37.02 | Osteomyelitis | 16 | 2.00 | Mild HF HL bilateral ears | Mild to moderate-severe sensorineural HL bilaterally | 14 |
| 6 | 40 | 21.0 | 124 | 17.23 | 14.20 | Osteomyelitis | 34 | 2.00 | Normal Hearing Bilaterally | Mild conductive HL left ear | 14 |
| 7 | 28 | 17.0 | 141.34 | 15.10 | 17.30 | Osteomyelitis | 40 | 2.00 | Normal Hearing bilaterally | Mild HL at HF bilaterally | 14 |

Abbreviations: Pt., patient; No., number; BMI, body mass index; CrCL, creatinine clearance; VAN, vancomycin; HL, hearing loss; LF, low frequency; HF, high frequency; Mid F, mid frequency; BL, bilateral

**Table 3. Bivariate analyses of variables associated with negative audiogram changes.**

| Variable | Crude Odds Ratio (95% CI) | *P*-value |
|---|---|---|
| Vancomycin ≥ 4 grams/day | 1.6 (0.3–8.8) | 0.63 |
| Vancomycin serum concentration ≥ 20 mcg/mL | 0.4 (0.1–2.1) | 0.4 |
| Age ≥ 65 years | 4.4 (0.5–38.2) | 0.24 |
| Concomitant ototoxins | 0.9 (0.9–1.0) | 0.6 |
| Osteoarticular infections | 5.8 (0.7–50.1) | 0.12 |
| Abnormal audiogram at baseline | 1.1 (0.22–4.9) | 1.0 |

four patients (57%) received concomitant antibiotics with vancomycin including cefepime, ertapenem and piperacillin-tazobactam. No patients received any concomitant ototoxins. Interestingly, no patients had vancomycin discontinued due to hearing loss, but rather for episodes of acute kidney injury (29%). The characteristics of patients with negative audiogram changes are listed in Table 2.

In bivariate analyses, no variables were found to be associated with a worsening change in audiogram, including a CART-derived age breakpoint of ≥ 65 years, elevated serum vancomycin levels, or vancomycin doses ≥ 4 grams/day (Table 3). No variables met criteria for inclusion in a multivariable regression model.

## Discussion

In this study, we found the overall prevalence of vancomycin-associated high frequency hearing loss to be low (8%). Most screened patients in our study had only a baseline audiogram

and did not present for outpatient follow-up testing; only 22% of patients had a baseline audiogram within 3 days of vancomycin initiation with weekly follow-up audiograms. The majority of audiogram changes were mild to moderate in nature, and only 3% of our patient cohort developed moderate to severe hearing loss while on vancomycin; two of the three patients were receiving concomitant antibiotics including piperacillin tazobactam, and ertapenem, and none received another ototoxin. Ototoxicity occurred at different times of vancomycin exposures with as early as two weeks in two patients and as long as 4 or 5 weeks in others. No variables were found to be associated with a worsening change in audiograms in bivariate analyses. The results of this study prompted our antimicrobial stewardship program to stop routine ordering of serial audiograms for patients receiving vancomycin, which we expect to have significant patient satisfaction and cost savings implications. The median cost of an audiogram is more than $300 United States dollars [18].

Contrary to the low incidence of ototoxicity observed in our study, Forouzesh et al. reported a 12% rate of high-frequency hearing loss in patients receiving vancomycin [17]. The mean of the highest vancomycin trough level in their cohort was 19 mcg/mL, which did not differ between patients who developed ototoxicity or did not. The investigators found that patients older than 53 years old were more likely to develop ototoxicity, but no other study has been published since to confirm these findings [17, 19]. In our study, patients greater than 65 years old were found to have an overall higher proportion of negative audiogram changes, but this was not proven to be a significant finding possibly due to an underpowered event or due to variability in the population age distribution found in our cohort. James and colleagues observed reversible ototoxicity following vancomycin concentrations >40 mcg/mL and irreversible damage with concentrations >80 mcg/mL or the presence of pre-existing renal impairment [20]. We found no association between elevated vancomycin concentrations or high (i.e., more than 4 grams per day) vancomycin daily doses in our cohort, which is representative of current vancomycin dosing strategies [17,19]. We also did not find impaired renal function to be a risk factor for ototoxicity.

Vancomycin associated ototoxicity is a rare complication; it is mainly cochlear and rarely significant and is usually reversible except in few reported cases [3,21]. There are published reports of infants overdosed with vancomycin without ill effects [22], and even vancomycin exposure in utero did not cause ototoxicity in neonates [23]. While ototoxicity has been more frequently reported in patients with underlying hearing problems or who receive concurrent ototoxic medications, no clear association between ototoxicity and vancomycin serum levels has been established [15,24]. Despite the low risk of vancomycin ototoxicity, it is recommended to discontinue vancomycin in patients experiencing signs of ototoxicity including tinnitus, loss of balance or loss of hearing [16].

Audiogram monitoring in the in-patient setting has many limitations. This primarily requires patient cooperation that can be complicated by the use of other medications which may affect cognitive function, or could also be impacted by the patient's underlying infection severity. Fifty-five percent of our patient population had an abnormal baseline audiogram, with noted improvement observed in some patients who received serial audiograms. Another limitation is that patients may have even a minor change of 2 decibels (dB) from baseline to follow-up audiogram and still meet the definition for a worsening audiogram. Campbell and colleagues reported that changes of > 5 dB are considered normal variability in an alert, oriented patient [25]. Alexander et al. performed large population-based cohort study evaluated the incidence of sudden sensorineural hearing loss in adults [26]. Results from that study suggested that the incidence of sudden sensorineural hearing loss (SSHL, unilateral loss of hearing occurring over 24 to 72 hours with 30 dB or more loss in at least 3 contiguous frequencies on pure-tone audiogram) was between 11 per 100,000 and 77 per 100,000 depending on the age

group evaluated, with patients over 65 years being at highest risk. These findings add to the difficulty of the interpreting repeated audiograms and establishing plausibility of vancomycin-associated ototoxicity, especially in scenarios where no follow up audiograms performed after stopping vancomycin.

Ototoxicity has important quality of life implications, and screening for ototoxicity is important in high-risk patients especially when receiving concomitant ototoxic medications. The use of rapid diagnostic techniques in clinical settings, such as polymerase chain reaction testing, can aid clinicians in promptly stopping vancomycin when not indicated and decrease unnecessary exposures [19].

Limitations to our study include that it was a small retrospective study with no randomization of the selected patients, and there were no follow up audiograms after completion of vancomycin treatment to assess reversibility in patients who developed ototoxicity. One other potential limitation was that audiograms were performed in different clinical settings (i.e., inpatient and outpatient), which may cause some variation in interpretation. The frequency of audiogram testing (i.e., weekly) was not protocolized or consistent among patients and may limit external validity, but we feel these are pragmatic data to reflect patient follow-up. Our study may not have been appropriately powered to detect risk factors for negative changes in audiograms, and did not assess vestibular toxicity. However, our data represent real-world practice of audiogram monitoring in what was perceived to be a high-risk population.

## Conclusions

The prevalence of high-level ototoxicity changes of audiograms in patients receiving long-term intravenous vancomycin was low. The utility of routine audiogram testing in this population does not seem warranted. However, larger studies may be warranted to further explore the risk of ototoxicity in high-risk patients and reversibility.

## Supporting information

**S1 File. Data analysis.**
(SAV)

## Author Contributions

**Conceptualization:** Mahmoud A. Shorman.

**Data curation:** Clayton Humphrey, Brian Walker.

**Formal analysis:** Michael P. Veve.

**Methodology:** Mahmoud A. Shorman.

**Project administration:** Mahmoud A. Shorman.

**Software:** Michael P. Veve.

**Writing – original draft:** Clayton Humphrey, Mahmoud A. Shorman.

**Writing – review & editing:** Michael P. Veve, Mahmoud A. Shorman.

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
