## [Decision Letter · Decision Letter 0]

10 Oct 2019

PONE-D-19-25571

Long-term vancomycin use had low risk of ototoxicity

PLOS ONE

Dear Dr Shorman,

Thank you for submitting your manuscript to PLOS ONE. After careful consideration, we feel that it has merit but does not fully meet PLOS ONE’s publication criteria as it currently stands. Therefore, we invite you to submit a revised version of the manuscript that addresses the points raised during the review process.

ACADEMIC EDITOR: 

In my opinion, authors should correct some missing data according to "Strobe Statement check list on cross-sectional studies": specifically:

point 9: Describe any efforts to address potential sources of bias;

point 10: Explain how the study size was arrived at; 

point 11:Explain how quantitative variables were handled in the analyses. If applicable, describe which groupings were chosen and why ;

point 12: (a) Describe all statistical methods, including those used to control for confounding (b) Describe any methods used to examine subgroups and interactions (c) Explain how missing data were addressed (d) If applicable, describe analytical methods taking account of sampling strategy  (e) Describe any sensitivity analyses.

 No conflicts between the reviews.

Please consider comments by reviewers.

We would appreciate receiving your revised manuscript by october 30th. To enhance the reproducibility of your results, we recommend that if applicable you deposit your laboratory protocols in protocols.io, where a protocol can be assigned its own identifier (DOI) such that it can be cited independently in the future. For instructions see: http://journals.plos.org/plosone/s/submission-guidelines#loc-laboratory-protocols

We look forward to receiving your revised manuscript.

Kind regards,

Martina Crivellari

Academic Editor

PLOS ONE

**Journal Requirements:**

2. In your ethics statement in the manuscript and in the online submission form, please provide additional information about the patient records used in your retrospective study. Specifically, please ensure that you have discussed whether all data were fully anonymized before you accessed them and/or whether the IRB or ethics committee waived the requirement for informed consent. If patients provided informed written consent to have data from their medical records used in research, please include this information.

**Comments to the Author**

1. Is the manuscript technically sound, and do the data support the conclusions?

Reviewer #1: Yes

Reviewer #2: Yes

Reviewer #3: Yes

2. Has the statistical analysis been performed appropriately and rigorously? 

Reviewer #1: No

Reviewer #2: Yes

Reviewer #3: Yes

3. Have the authors made all data underlying the findings in their manuscript fully available?

Reviewer #1: Yes

Reviewer #2: Yes

Reviewer #3: Yes

4. Is the manuscript presented in an intelligible fashion and written in standard English?

Reviewer #1: Yes

Reviewer #2: Yes

Reviewer #3: Yes

5. Review Comments to the Author

Reviewer #1: Dear Dr Martina Crivellari,

Re: PONE-D-19-25571 Long-term vancomycin use had low risk of ototoxicity

Thank you for the kind invitation to review the manuscript. This study addresses an important evidence gap on vancomycin related ototoxicity. Below are my comments to the authors as there are some areas that need to be addressed before the manuscript can be published.

General comments:

Language

There are some incomplete sentences and grammatical errors scattered within the manuscript. I would suggest for the authors to re-read their manuscript prior to the next submission and correct the necessary errors.

Main text:

Methods:

1) Given the inter-physician variability on usage of audiogram for monitoring of vancomycin related ototoxicity, are there general principles that the clinicians in your centre apply when selecting patients to undergo audiogram? It would be worthwhile to describe this in the methodology and also the uptake rate of using audiogram for patients among physicians in your institution.

2) What is the change in audiogram findings that is deemed as significant in the study? This needs to be defined in the methodology for clinical interpretability.

Results:

1) A significant proportion of patients were excluded as audiogram was not performed. Were the demographics of patients included representative or comparable with entire patient population on vancomycin?

2) p-value for table 1 should be reported as the authors mentioned that univariate analyses was performed in their methodology.

3) Results for the renal function of patients should be segregated by CKD stage or eGFR cut-offs for readers to have a better understanding on renal profiles of included patients.

4) For Table 2, it would be useful to know when the audiogram abnormalities were detected (during their course of vancomycin therapy) and also the specific concomitant ototoxic agents that the patients may be on. If the salient points of audiogram findings can be presented in a weekly basis, it would be able to add value in terms of points for discussion on the onset of vancomycin ototoxicity among your patient population. I am also interested to know if vancomycin therapy was stopped following detection of audiogram abnormalities.

5) For the bivariate analyses, how was the cut-off of 65 years old selected? I noted the study by Forouzesh et al (also cited by the authors) identified the age cut-off of 53 years old to be a risk factor for ototoxicity. The authors may wish to contemplate using that cut-off instead. I also did notice that there is some discrepancies in the age cut-off reported in Table 3 (age >=40 years old) and in the main text (age >65 years old). Do clarify which cutoff was used in the analyses.

6) It is intriguing that among patients who developed audiogram abnormalities, some get 2 audiograms while others get up to 7 audiograms. What I would like to understand is how the frequency of monitoring for audiogram was first derived and determined within your institution?

Discussion

1) The first 2 paragraphs of the discussion appear to be less relevant to the topic that this manuscript wishes to cover. Do consider removing it.

2) Page 10 writes that 424 patients with weekly audiogram orders were evaluated in the study due to perceived vancomycin ototoxicity. Do the authors mean that these patients reported some hearing deficits?

3) Do the authors have any postulated reasons why age was not found to be a risk factor for vancomycin related ototoxicity?

4) The definition of changes in audiogram findings deemed to be significant should be described in the methodology instead of discussion.

5) Do the authors have any details on the baseline characteristics of their patients pertaining to dementia or other cognitive health problems which may confound their ability to comply with instructions for audiogram assessment?

Others minor comments

1) The keywords selected for the manuscript needs to be amended. Do use Mesh Terms where appropriate to allow wider reach of the article. I believe that the current keywords reflect the objective of the study instead of the keywords

Introduction:

2) Page 4 Line 87 to 90 – pls consider citing International journal of clinical pharmacy 40 (5), 977-981

3) There are some grammatical errors in the section for authors’ contributions. Kindly amend it.

4) There are some spelling errors in the manuscript. E.g. Page 8 Line 172

Reviewer #2: This is an original work which tries to find out whether vancomycin causes significant ototoxicity in long term. The patients size is somewhat sufficient and audiometric evaluations has been performed before and after or during the treatment. I think, this study merits to be published, because it provides a clear information about the safety of vancomycin in terms of ototoxicity.

I have an easy advice:

I recommend to use always ‘estimated glomerular filtration rate (eGFR)’ instead of ‘estimated creatinine clearance’. Probably, they used CKD-EPI equation for it. So it maybe it would be better to cite it (Ann Int Med 2009; 150(9):604-612).

In the abstract

Line 47, Line 153 and in Table 1.

Reviewer #3: The study presents the results of original research and the authors reported that the results have not been published elsewhere. The statistics analysis was performed with technical standard and is described in sufficient detail, although it is a simple analysis, it fulfills the purpose set by the authors. The conclusions presented are supported by the data; according to the objective of the study, the prevalence of negative changes in audiograms of the patients in their medical center was determined; however, the analysis was carried out with very small sample size. Since there are few reports that take into account the hearing damage associated to prolonged use of vancomycin, in an objective way, it can be considered that the data provided by this study could serve as a guide for other medical centers.

It is recommended that the authors review the document, to correct errors regarding the use of vancomycin concentration units, as well as the correct use of the abbreviation et al. There are errors in the writing of references, they are not uniform. Except for minor changes, it can be considered accepted, taking into account that the brief results could serve as a guide for other medical centers.

6. PLOS authors have the option to publish the peer review history of their article (what does this mean?). If published, this will include your full peer review and any attached files.

Reviewer #1: No

Reviewer #2: No

Reviewer #3: No

---

## [Author Response · Author response to Decision Letter 0]

14 Oct 2019

Academic Editor

Comment 1. Describe any efforts to address potential sources of bias

Response 1. We have addressed potential sources of bias of confounding in the discussion section of our manuscript (lines 268-278).

Comment 2. Explain how the study size was arrived at; 

Response 2. We have added sample size information in the manuscript (lines 147-148)

“Given the exploratory nature of this project, no formal sample size calculations were performed.”

Comment 3. Explain how quantitative variables were handled in the analyses. If applicable, describe which groupings were chosen and why

Response 3. We have described our statistical methods in lines 147-157.

Comment 4. (a) Describe all statistical methods, including those used to control for confounding (b) Describe any methods used to examine subgroups and interactions (c) Explain how missing data were addressed (d) If applicable, describe analytical methods taking account of sampling strategy (e) Describe any sensitivity analyses.

Response 4. We have described our statistical methods in lines 147-157.

Reviewer #1

Comment 1. There are some incomplete sentences and grammatical errors scattered within the manuscript. I would suggest for the authors to re-read their manuscript prior to the next submission and correct the necessary errors.

Response 1. Thank you for this comment. We have critically reviewed the paper for grammar errors and misspellings. We believe we have addressed all of these issues. 

Comment 2. Given the inter-physician variability on usage of audiogram for monitoring of vancomycin related ototoxicity, are there general principles that the clinicians in your centre apply when selecting patients to undergo audiogram? It would be worthwhile to describe this in the methodology and also the uptake rate of using audiogram for patients among physicians in your institution.

Response 2. We appreciate this insight. No formal principles were used to select patients under-going audiograms, outside of long-term vancomycin use, and this process was limited to the discretion of the ordering physician. We have clarified this throughout the manuscript (lines 101-104):

“Given the perceived risks of vancomycin- associated ototoxicity and the commonality of invasive MRSA infections in our region, audiogram monitoring for patients receiving prolonged vancomycin for any indication was adapted by select infectious diseases physicians at our institution.”

And (lines 136-139)

“The procedure for audiogram monitoring in patients receiving vancomycin was not formally protocolized, but patients thought to be at risk for ototoxicity by the ordering provider typically received orders for weekly audiograms throughout the duration of vancomycin.”

Comment 3. What is the change in audiogram findings that is deemed as significant in the study? This needs to be defined in the methodology for clinical interpretability.

Response 3. We have included the interpretive audiogram results that coincide with any significant changes as interpreted by certified, hospital-based audiologists (lines 139-145):

“Baseline and follow-up audiogram results were collected; certified audiologists interpreted audiograms. Findings were recorded for both ears ranked in order of increasing severity based on the sound intensity required for hearing as follows: normal, <25 dB; mild, 25 to 40 dB; moderate, 40 to 70 dB; severe, 70 to 90 dB; profound, >90 dB sensorineural hearing loss. Patients were considered to have a negative change in audiogram if there was a noted increase in the sound intensity required for hearing and worsening of more than 5 db from baseline.”

Comment 4. A significant proportion of patients were excluded as audiogram was not performed. Were the demographics of patients included representative or comparable with entire patient population on vancomycin?

Response 4. We did collect sufficient demographic data on patients who did not receive at least 14 days of vancomycin, or who did not receive at least two follow-up audiograms. However, we have no reason to suspect the population of patients included in this study are not representative of the general population of patients receiving vancomycin. We unfortunately do not have IRB approval to access data on patients who were excluded from this study.

Comment 5. p-value for table 1 should be reported as the authors mentioned that univariate analyses was performed in their methodology.

Response 5. We have included P-values for the data included in Table 1.

Comment 6. Results for the renal function of patients should be segregated by CKD stage or eGFR cut-offs for readers to have a better understanding on renal profiles of included patients.

Response 6. Thank you for this comment. We measured renal function using the Cockcroft-Gault equation, and this has been updated throughout the manuscript when creatinine clearance is discussed. We have also segregated patient renal function into different cut-offs as suggested, in Table 1.

Comment 7. For Table 2, it would be useful to know when the audiogram abnormalities were detected (during their course of vancomycin therapy) and also the specific concomitant ototoxic agents that the patients may be on. If the salient points of audiogram findings can be presented in a weekly basis, it would be able to add value in terms of points for discussion on the onset of vancomycin ototoxicity among your patient population. I am also interested to know if vancomycin therapy was stopped following detection of audiogram abnormalities.

Response 7. Thank you for this comment. None of the patients who developed a negative change in audiogram were on a concomitant ototoxin (line 189-190, Table 2), and no patients discontinued vancomycin secondary to hearing loss (line 190-191). In regards to timing of negative audiogram change, we have added this information in the results section of the manuscript (line 187) and in Table 2.

Comment 8. For the bivariate analyses, how was the cut-off of 65 years old selected? I noted the study by Forouzesh et al (also cited by the authors) identified the age cut-off of 53 years old to be a risk factor for ototoxicity. The authors may wish to contemplate using that cut-off instead. I also did notice that there is some discrepancies in the age cut-off reported in Table 3 (age >=40 years old) and in the main text (age >65 years old). Do clarify which cutoff was used in the analyses.

Response 8. Thank you for this observation. The cut-off of 65 years was derived from a classification and regression tree (CART) analyses, as described in our methods (lines 150-152):

“Classification and regression tree (CART) analyses were performed to identify dichotomous breakpoints in continuous variables associated with a negative change in audiogram.”

The breakpoint of 65-years is based only on our population distribution, as was 53-years determined by Forouzesh and colleagues. We feel using 65-years as a cut-off is a better representation of our own data as opposed to Forouzesh and colleagues. We have also updated Table 3 to state 65-years instead of 40 years. Thank you for identifying this oversight. 

Comment 9. It is intriguing that among patients who developed audiogram abnormalities, some get 2 audiograms while others get up to 7 audiograms. What I would like to understand is how the frequency of monitoring for audiogram was first derived and determined within your institution?

Response 9. Thank you for this comment. While there is no standardized process in place, the general approach was to obtain weekly audiograms while the patient was on vancomycin. This was included in the manuscript in lines 136-139: 

“The procedure for audiogram monitoring at our institution was not formally protocolized, but patients thought to be at risk for ototoxicity by the ordering provider typically received orders for weekly audiograms throughout the duration of vancomycin.”

And (lines 273-275)

“The frequency of audiogram testing (i.e., weekly) was not protocolized or consistent among patients and may limit external validity, but we feel these are pragmatic data to reflect patient follow-up.”

Comment 10. The first 2 paragraphs of the discussion appear to be less relevant to the topic that this manuscript wishes to cover. Do consider removing it.

Response 10. Thank you for this feedback. We have removed these paragraphs from the discussion. 

Comment 11. Page 10 writes that 424 patients with weekly audiogram orders were evaluated in the study due to perceived vancomycin ototoxicity. Do the authors mean that these patients reported some hearing deficits?

Response 11. We originally generated a screening list based on the following criteria: age > 18 years, receipt of vancomycin order, and receipt of at least one audiogram or an audiogram order placed in the electronic health record. We then proceeded to screen the 424 patients to assess if they met all inclusion criteria. 

We have added the following sentence for clarity (lines 160-162): 

“A list of adult patients who received vancomycin and at least one audiogram or audiogram order placed in the electronic health record was created in order to further screen patients for inclusion.”

Comment 12. Do the authors have any postulated reasons why age was not found to be a risk factor for vancomycin related ototoxicity?

Response 12. We hypothesize that our study may have been underpowered to detect other variables associated with ototoxicity when long-term vancomycin is used. We have included this rationale in the discussion, lines 221-225: 

“In our study, patients greater than 65 years old were found to have an overall higher proportion of negative audiogram changes, but this was not proven to be a significant finding possibly due to an underpowered event or due to variability in the population age distribution found in our cohort.”

Comment 13. The definition of changes in audiogram findings deemed to be significant should be described in the methodology instead of discussion.

Response 13. We have included these data in the methods section of the manuscript (lines 140-145). Our comment in the discussion was only to highlight that a small decibel difference in would lead potentially lead significant changes in audiogram interpretation, and is an inherent limitation of our findings.

Comment 14. Do the authors have any details on the baseline characteristics of their patients pertaining to dementia or other cognitive health problems which may confound their ability to comply with instructions for audiogram assessment?

Response 14. We do not have specific data on dementia diagnoses or baseline impaired cognition in from electronic health record data. However, our audiologists screen and comment on any potential confounding factors that may skew the results of the audiogram test. None of the patients included would have received an audiogram if they were deemed to not have the capacity to reliably comply to testing, based on institutional standards. We have therefore added language that patients with dementia or cognitive disorders were excluded from the study (lines 123-125): 

“Patients were excluded if they had dementia or other cognitive dysfunction, a documented history of perforated tympanic membrane or the presence of hearing aids or cochlear implants, or if they did not have any follow-up audiograms.”

Comment 15. The keywords selected for the manuscript needs to be amended. Do use Mesh Terms where appropriate to allow wider reach of the article. I believe that the current keywords reflect the objective of the study instead of the keywords

Response 15. We have modified the article keywords to MeSH Terms: vancomycin, hearing loss, drug toxicity, antimicrobial stewardship, methicillin-resistant Staphylococcus aureus

Comment 16. Page 4 Line 87 to 90 – pls consider citing International journal of clinical pharmacy 40 (5), 977-981

Response 16. Thank you for this feedback. We have incorporated this publication from Seng and colleagues to support our statement related to vancomycin ototoxicity.

Comment 17. There are some grammatical errors in the section for authors’ contributions. Kindly amend it.

Response 17. We have modified the grammatical errors for the Author’s Contributions (lines 13-17): 

“CH established the electronic case report form, performed data collection, and assisted in manuscript preparation. MPV performed the statistical analyses and assisted in manuscript preparation. BW assisted in establishing the electronic case report form and performed data collection. MAS designed the study and wrote the manuscript.”

Comment 18. There are some spelling errors in the manuscript. E.g. Page 8 Line 172

Response 18. We have thoroughly reviewed the manuscript in the context of spelling and grammar. Thank you for this comment.

Reviewer #2

Comment 1. I have an easy advice:

I recommend to use always ‘estimated glomerular filtration rate (eGFR)’ instead of ‘estimated creatinine clearance’. Probably, they used CKD-EPI equation for it. So it maybe it would be better to cite it (Ann Int Med 2009; 150(9):604-612).

In the abstract

Line 47, Line 153 and in Table 1.

Response 1. Thank you for this comment. The actual equation used to assess renal function was the Cockcroft-Gault equation, and not eGFR. We have corrected this error throughout the manuscript and in Table 1. Furthermore, we have stratified the creatinine clearance by range instead of reporting the median (IQR), as suggested by Reviewer #1.

Reviewer #3

Comment 1. It is recommended that the authors review the document, to correct errors regarding the use of vancomycin concentration units, as well as the correct use of the abbreviation et al. 

Response 1. Thank you for these comments. To stay consistent with vancomycin unit reporting, all units have been modified to micrograms per milliliter (mcg/mL). Additionally, we have made appropriate modifications regarding the correct use of et al., where applicable.

Comment 2. There are errors in the writing of references, they are not uniform. 

Response 2. We have modified the reference formats per journal instructions.

---

## [Editor Report · Decision Letter 1]

17 Oct 2019

Long-term vancomycin use had low risk of ototoxicity

PONE-D-19-25571R1

Dear Dr. Shorman,

We are pleased to inform you that your manuscript has been judged scientifically suitable for publication and will be formally accepted for publication once it complies with all outstanding technical requirements.

With kind regards,

Martina Crivellari

Academic Editor

PLOS ONE

Additional Editor Comments (optional):

Reviewers' comments:

All comments have been addressed.

---

## [Editor Report · Acceptance letter]

28 Oct 2019

PONE-D-19-25571R1 

Long-term vancomycin use had low risk of ototoxicity 

Dear Dr. Shorman:

I am pleased to inform you that your manuscript has been deemed suitable for publication in PLOS ONE. Congratulations! Your manuscript is now with our production department. 

With kind regards,

on behalf of

Dr. Martina Crivellari 

Academic Editor

PLOS ONE